# Diffeomorphism Covariance of the Canonical Barbero–Immirzi–Holst Triad Theory

Donald Salisbury 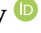

Austin College, 900 N. Grand Ave., Sherman, TX 75090, USA; dsalisbury@austincollege.edu

**Abstract:** The vanishing phase space generator of the full four-dimensional diffeomorphism-related symmetry group in the context of the Barbero–Immirz–Holst Lagrangian is derived directly, for the first time, from Noether's second theorem. Its applicability in the construction of classical diffeomorphism invariants is reviewed.

**Keywords:** diffeomorphism symmetry; gravitational invariants; classical loop variables





## 1. Introduction

What we identify as the Barbero–Immirzi–Holst model serves as a foundation for today's canonical approach to loop quantum gravity. In this article, we derive a new analysis of the underlying four-dimensional spacetime diffeomorphism-related classical canonical symmetry. We derive the canonical symmetry generators directly from the vanishing charge that follows from Emmy Noether's second theorem, in a manner similar to the first such derivation presented for conventional canonical gravity in [1]. The focus is on a reformulated ADM approach that incorporates densited triads. And we argue that the extension of this analysis to the new triad approach to gravity as proposed in [2–4] is almost trivial. As is well known, in order to achieve the results of canonically generated variations of spacetime coordinates, it is necessary to supplement the variations of phase space variables under diffeomorphims with related triad gauge transformations. We conclude with an overview of a technique for introducing intrinsic coordinates as gauge conditions and employing the full diffeomorphism generator to construct invariant temporal evolution in a manner related to Rovelli's relative observables [5]. This lays the foundations for an eventual application in loop quantum gravity.

It should be stressed that this underlying phase space approach to symmetry is still not widely appreciated in the general relativity community. We have written much on the historical background of the dispute amongst proponents of the original Wheeler–Dewitt formalism and approaches developed by Peter Bergmann, including many collaborators and students. See, for example, [6]. The key distinction in this dispute is Wheeler's abandonment of the full spacetime diffeomorphism symmetry. He did retain a three-dimensional phase space covariance, but he abandoned the full spacetime symmetry in a phase space framework in favor of a notion that he called 'multi-fingered time' [7]. At least intially he thought that the spatial metric itself could fix the time evolution. The views have evolved substantially, and in fact today there is a much wider appreciation of the potential to implement the full four-dimensional diffeomorphism symmetry in the general relativistic phase space. Indeed, several authors have addressed the question of how one can invoke this symmetry in constructing true spacetime diffeomorphism invariants. Especially noteworthy are current efforts to employ the full symmetry, recognizing that the full general relativistic metric, including the so-called lapse and shift functions, must be included as phase space functions. We cite, in particular, the recent work in quantum cosmology [8–12] where intrinsic coordinates are being employed in the context of quantum loop cosmology. The coordinates themselves are defined in terms of either of the spacetime metric or material

fields. One can thereby in principal employ the diffeomorphism symmetry group that we are deriving in a new manner in this article to construct objects that are invariant under the action of the group. This is closely related to an approach called relational quantum mechanics that has long been advocated by Rovelli [13].

## 2. Derivation of Canonical Hamiltonian

We use minus one half of the ADM Lagrangian as rewritten using triad variables.

$$\mathcal{L} = -\frac{1}{2}\mathcal{L}_{ADM} = -\frac{1}{2}Nt\left({}^{3}R + K_{ab}K^{ab} - (K_a^a)^2\right) = -\frac{1}{2}Nt\left({}^{3}R + K_{ab}e^{ac}e^{bd}K_{cd} - \left(e^{ab}K_{ab}\right)^2\right), \tag{1}$$

where

$$K_{ab} = \frac{1}{2N}\left(g_{ab,0} - N^c g_{ab,c} - g_{ca}N^c_{,b} - g_{cb}N^c_{,a}\right) = \frac{1}{2N}\left(g_{ab,0} - 2g_{c(a}N^c_{|b)}\right). \tag{2}$$

The variable $t$ is the determinant of the spatial metic $g_{ab}$, with $e^{ab}$ its inverse. The variable $N$ is the lapse while $N^a$ represents the metric shift functions. ${}^3R$ is the tree-dimensional curvature scalar.

The first task is to specialize to tetrads with the choice $E_0^\mu = n^\mu = \delta_0^\mu N^{-1} - \delta_a^\mu N^{-1}N^a$. This tetrad is orthogonal to the constant time hypersurface. The covariant metric is

$$g_{\mu\nu} = \begin{pmatrix} -N^2 + N^c N^d g_{cd} & g_{ac}N^c \\ g_{bd}N^d & g_{ab} \end{pmatrix}, \tag{3}$$

with the contravariant metric

$$g^{\mu\nu} = \begin{pmatrix} -1/N^2 & N^a/N^2 \\ N^b/N^2 & e^{ab} - N^a N^b/N^2 \end{pmatrix}. \tag{4}$$

We then choose the remaining tetrads to be tangential to the constant time hypersurface. Thus, the full set of contravariant tetrads (with the upper index representing the row and the lower index representing the column) is

$$E_I^\mu = \begin{pmatrix} N^{-1} & 0 \\ -N^{-1}N^a & T_i^a \end{pmatrix}, \tag{5}$$

with the corresponding covariant set

$$e_\mu^I = \begin{pmatrix} N & 0 \\ t_a^i N^a & t_a^i \end{pmatrix}. \tag{6}$$

However, we employ as independent triad variables $\tilde{T}_i^a := tT_i^a$ where $t := \det(t_a^i)$. Furthermore, rather than choosing the lapse $N$ as an independent configuration variable, we work with $\underset{\sim}{N} := t^{-1}N$. Therefore, for the following, we need

$$t_{,\mu} = tt_{a,\mu}^i T_i^a = \left(t_a^i \tilde{T}_i^a\right)_{,\mu} - t_a^i \tilde{T}_{i,\mu}^a; \tag{7}$$

therefore, we find that

$$t_{,\mu} = \frac{1}{2}t_a^i \tilde{T}_{i,\mu}^a, \tag{8}$$

$$t_{a,\mu}^i = t^{-1}\tilde{T}_{j,\mu}^b\left(-t_b^i t_a^j + \frac{1}{2}t_b^j t_a^i\right), \tag{9}$$

and

$$T_{i,\mu}^a = -\frac{1}{2}t^{-2}t_b^j \tilde{T}_{j,\mu}^b \tilde{T}_i^a + t^{-1}\tilde{T}_{i,\mu}^a. \tag{10}$$

Now, we define the canonical momentum

$$
\begin{aligned}
p_e^l \; &:= \; \frac{\partial \mathcal{L}}{\partial \tilde{T}_{l,0}^e} \\
&= \; -Nt\left(e^{ac}e^{bd} - e^{ab}e^{cd}\right)K_{cd}\frac{\partial K_{ab}}{\partial \tilde{T}_{l,0}^e}.
\end{aligned}
\tag{11}
$$

Therefore, we need

$$
2Nt\frac{\partial K_{ab}}{\partial \tilde{T}_{l,0}^e} = g_{ab}t_e^l - 2t_{(a}^l g_{b)e}.
\tag{12}
$$

Subsequently,

$$
p_e^l = T_l^d K_{ed},
\tag{13}
$$

from which we deduce that

$$
p_{(a}^i t_{b)}^i = K_{ab}.
\tag{14}
$$

Therefore, we can write the Lagrangian immediately in terms of the canonical momenta.

To obtain the canonical Hamiltonian $\mathcal{H}_c$, we must now focus on $p_a^i \tilde{T}_{i,0}^a$ which we want to write in terms of the momenta. We have

$$
p_a^i \tilde{T}_{i,0}^a = K_{ab}T_i^b \tilde{T}_{i,0}^a.
\tag{15}
$$

We rewrite this in terms of derivatives of $t_c^j$. Therefore, we consider first

$$
\tilde{T}_{i,0}^a = (tT_i^a)_{,0} = t_{,0}T_i^a + tT_{i,0}^a = tt_{c,0}^j T_j^c T_i^a - tT_i^c T_j^a t_{c,0}^j,
\tag{16}
$$

and we therefore have

$$
p_a^i \tilde{T}_{i,0}^a = \frac{1}{2}tK_{ab}\left(e^{ab}e^{cd} - e^{bc}e^{ad}\right)g_{cd,0}.
\tag{17}
$$

But

$$
g_{cd,0} = 2NK_{cd} + 2g_{e(c}N_{|d)}^e,
\tag{18}
$$

so we conclude, finally, that

$$
p_a^i \tilde{T}_{i,0}^a = tK_{ab}\left(e^{ab}e^{cd} - e^{bc}e^{ad}\right)\left(NK_{cd} + g_{e(c}N_{|d)}^e\right).
\tag{19}
$$

We thereby obtain the expression for the canonical Hamiltonian,

$$
\begin{aligned}
\mathcal{H}_c \; &= \; p_a^i \tilde{T}_{i,0}^a - \mathcal{L} \\
&= \; \frac{Nt}{2}\left({}^3R + K_{ab}e^{ac}e^{bd}K_{cd} - \left(e^{ab}K_{ab}\right)^2\right) + tK_{ab}\left(e^{ab}N_{|c}^c - e^{ac}N_{|c}^b\right).
\end{aligned}
\tag{20}
$$

For later use, we need to rewrite the canonical Hamiltonian in terms of $p_a^i$ using $K_{ab} = p_{(a}^i t_{b)}^i$, which implies that

$$
K_{ab}e^{ac}e^{bd}K_{cd} = p_{(a}^i t_{b)}^i p_{(c}^j t_{d)}^j e^{ac}e^{bd} = \frac{1}{2}\left(p_a^i p_b^i e^{ab} + p_a^i p_b^j T_j^a T_i^b\right)
\tag{21}
$$

and

$$
e^{ab}K_{ab}e^{cd}K_{cd} = p_a^i T_i^a p_b^j T_j^b.
\tag{22}
$$

Therefore, the canonical Hamitonian becomes

$$
\begin{aligned}
\mathcal{H}_c &= \frac{Nt}{2}\left({}^3R + \frac{1}{2}p_a^i p_b^i e^{ab} + \frac{1}{2}p_a^i p_b^j T_j^a T_i^b - p_a^i T_i^a p_b^j T_j^b\right) + t p_a^i t_b^i\left(e^{ab} N_{|c}^c - e^{c(a} N_{|c}^{b)}\right) \\
&= \frac{N}{\underset{\sim}{2}}\left(\tilde{T}_i^a \tilde{T}_j^b {}^3R_{ab}^{ij} + \frac{1}{2}p_a^i p_b^i \tilde{T}_j^a \tilde{T}_j^b + \frac{1}{2}p_a^i p_b^j \tilde{T}_j^a \tilde{T}_i^b - p_a^i \tilde{T}_i^a p_b^j \tilde{T}_j^b\right) \\
&\quad + t p_a^i t_b^i\left(e^{ab} N_{|c}^c - e^{c(a} N_{|c}^{b)}\right) \\
&= \underset{\sim}{N}\tilde{\tilde{\mathcal{H}}}_0 + p_a^i \tilde{T}_i^a N_{|c}^c - \frac{1}{2}p_a^i \underset{\sim}{t}_b^i \tilde{T}_j^c \tilde{T}_j^a N_{|c}^b - \frac{1}{2}p_b^i \tilde{T}_i^c N_{|c}^b,
\end{aligned}
\tag{23}
$$

where $\underset{\sim}{N} := t^{-1}N$ and

$$
\tilde{\tilde{\mathcal{H}}}_0 := -\frac{1}{2}\tilde{T}_i^a \tilde{T}_j^b\left({}^3R_{ab}^{ij} + p_a^i p_b^i - p_b^i p_a^j\right).
\tag{24}
$$

It is straightforward to check that this does deliver an almost correct expression for the time rate of change in the densitized triad—lacking, as we shall see shortly, the arbitrary triad gauge rotations, i.e.,

$$
\tilde{T}_{l,0}^e = \frac{\partial \mathcal{H}_c}{\partial p_e^l} = \underset{\sim}{N}\left(p_b^l \tilde{T}_j^e \tilde{T}_j^b - p_b^j \tilde{T}_j^b \tilde{T}_l^e\right) + \tilde{T}_l^e N_{|c}^c - \frac{1}{2}\underset{\sim}{t}_b^l \tilde{T}_j^c \tilde{T}_j^e N_{|c}^b - \frac{1}{2}\tilde{T}_l^c N_{|c}^e.
\tag{25}
$$

It is important to recognize here that the ADM Lagrangian does not depend on the antisymmetrized linear combination of velocities $\tilde{T}^{a[i}\underset{\sim}{t}_a^{j]}$, and as a consequence, we obtain a corresponding primary constraint, with a corresponding addition to the Hamiltonian generator of time evolution. Rosenfeld, indeed, in [14] considered a tetrad version of general relativity in which analogous constraints appeared and, although he did not explicitly construct the corresponding extended Hamiltonian, he could easily have applied his new techniques to achieve this. We next derive the relevant primary constraint by applying Noether's second theorem.

### 3. Noether Charges

First, there is a vanishing charge that arises from the invariance of the ADM action under triad rotations

$$
\delta_\eta T_i^a = \epsilon^{ijk}\tilde{T}_j^a \eta_k,
\tag{26}
$$

where $\eta_k$ are arbitrary spacetime functions. Following Noether's second theorem, conserved charge arises as follows. The variation of the action is

$$
0 = \delta_\eta \int d^4x\,\mathcal{L} = \int d^4x\left[\left(\frac{\delta\mathcal{L}}{\delta\tilde{T}_i^a}\right)\delta_\eta \tilde{T}_i^a + \left(\frac{\partial\mathcal{L}}{\partial\tilde{T}_{j,\mu}^a}\epsilon^{ijk}\tilde{T}_j^a \eta_k\right)_{,\mu}\right].
\tag{27}
$$

When the field equations are satisfied, we obtain, letting the variations vanish at spatial infinity, the conserved charge

$$
C_\eta = \int d^3x\, p_a^i \epsilon^{ijk}\tilde{T}_j^a \eta_k.
\tag{28}
$$

But since $\eta_k$ can vary arbitrarily with time, we deduce the existence of constraints

$$
0 = \tilde{\mathcal{H}}^k := -\epsilon^{ijk}p_a^i \tilde{T}_j^a.
\tag{29}
$$

The additional constraints that arise from the invariance of the action under spacetime diffeomorphisms require a bit more work to derive. We derive the vanishing Noether charge diffeomorphism-related generator following the procedure that was applied in the

conventional metric case in [1]. It should be noted here that this procedure was applied to tetrad-based general relativity by Rosenfeld in 1930. And as observed in [15], he did not complete the derivation of the canonical generators that we shortly find, very likely because he recognized that he could not express them exclusively in terms of canonical variables. In other words, he did not recognize, as first observed in [16], that the variations were not projectable under the Legendre transformation to phase space.

Under an infinitesimal diffeomorphism $x'^\mu = x^\mu - \epsilon^\mu$, the scalar density $\mathcal{L}$ transforms as [1]

$$\bar{\delta}\mathcal{L} = (\mathcal{L}\epsilon^\mu)_{,\mu}, \tag{30}$$

where the $\bar{\delta}$ variation is actually the Lie derivative $\mathcal{L}_\epsilon$. We shortly work out the corresponding field variations. But first, we derive the corresponding vanishing Noether charges noting that when the field equations are satisfied, and letting $\epsilon^a \to 0$ at spatial infinity,

$$
\begin{aligned}
\int d^4x\, \bar{\delta}\mathcal{L} &= \int d^3x \left( \frac{\partial \mathcal{L}}{\partial \tilde{T}^a_{i,0}} \bar{\delta}\tilde{T}^a_i + \frac{\partial \mathcal{L}}{\partial N_{,0}} \bar{\delta}N + \frac{\partial \mathcal{L}}{\partial N^a_{,0}} \bar{\delta}N^a \right) \Bigg|_{x^0_i}^{x^0_f} \\
&= \int d^3x\, \mathcal{L}\epsilon^0 \Big|_{x^0_i}^{x^0_f}.
\end{aligned}
\tag{31}
$$

Therefore, again, taking into account that the time dependence of $\epsilon^\mu$ is arbitrary, we derive the corresponding vanishing Noether charges

$$C_\epsilon = \int d^3x\, \mathfrak{C}_\epsilon \tag{32}$$

with vanishing charge density

$$
\begin{aligned}
\mathfrak{C}_\epsilon &= \frac{\partial \mathcal{L}}{\partial \tilde{T}^a_{i,0}} \bar{\delta}\tilde{T}^a_i + \frac{\partial \mathcal{L}}{\partial N_{,0}} \bar{\delta}N + \frac{\partial \mathcal{L}}{\partial N^a_{,0}} \bar{\delta}N^a - \mathcal{L}\epsilon^0 \\
&= p^i_a \bar{\delta}\tilde{T}^a_i + \tilde{\tilde{P}}\bar{\delta}N + \tilde{P}_a \bar{\delta}N^a - \mathcal{L}\epsilon^0.
\end{aligned}
\tag{33}
$$

We recognize, of course, that the momenta $\tilde{\tilde{P}}$ and $\tilde{P}_a$ are primary constraints.

The next step is to determine the variations under $x'^\mu = x^\mu - \epsilon^\mu$. We must bear in mind that the variations of the triads must yield vectors that remain tangent to the fixed time hypersurface. And furthermore, the varied $n^\mu = \delta^\mu_0 N^{-1} - \delta^\mu_a N^{-1}N^a$ must be perpendicular to this new hypersurface. The resulting variations are

$$\bar{\delta}N = N\epsilon^0_{,0} - NN^a\epsilon^0_{,a} + N\epsilon^0_{,0} + N_{,a}\epsilon^a \tag{34}$$

and

$$\bar{\delta}N^a = N^a\epsilon^0_{,0} - (N^2 e^{ab} + N^a N^b)\epsilon^0_{,b} + \epsilon^a_{,0} - N^b\epsilon^a_{,b} + N^a_{,0}\epsilon^0 + N^a_{,b}\epsilon^b. \tag{35}$$

To determine the variation of $\tilde{T}^a_i$, we refer to the variation of the spatial components of the metric. We have

$$
\begin{aligned}
\bar{\delta}g_{ab} &= \bar{\delta}t^i_a t^i_b + t^i_a \bar{\delta}t^i_b \\
&= t^i_{a,\mu}\epsilon^\mu t^i_b + t^i_a t^i_{b,\mu}\epsilon^\mu + t^i_c N^c \epsilon^\mu_{,a} t^i_b + t^i_c \epsilon^c_{,a} t^i_b + t^i_a t^i_c N^c \epsilon^0_{,b} + t^i_a t^i_c \epsilon^c_{,b}.
\end{aligned}
\tag{36}
$$

Therefore, we find

$$\bar{\delta}t^i_a = t^i_{a,\mu}\epsilon^\mu + t^i_b N^b \epsilon^0_{,a} + t^i_b \epsilon^b_{,a}. \tag{37}$$

Next, we calculate $\bar{\delta} T_i^a$ using

$$\bar{\delta} t_a^i T_j^a = -t_a^i \bar{\delta} T_j^a, \tag{38}$$

which implies

$$
\begin{aligned}
\bar{\delta} T_j^b &= -\bar{\delta} t_a^i T_j^a T_i^b = -\left( t_{a,\mu}^i \epsilon^\mu + t_c^i N^c \epsilon_{,a}^0 + t_c^i \epsilon_{,a}^c \right) T_j^a T_i^b \\
&= T_{j,\mu}^b \epsilon^\mu - N^b T_j^a \epsilon_{,a}^0 - \epsilon_{,a}^b T_j^a.
\end{aligned} \tag{39}
$$

Now, to obtain $\bar{\delta} \tilde{T}_i^a$, we need

$$\bar{\delta} t = t \bar{\delta} t_a^i T_i^a = t \left( t_{a,\mu}^i \epsilon^\mu + t_b^i N^b \epsilon_{,a}^0 + t_b^i \epsilon_{,a}^b \right) T_i^a, \tag{40}$$

which implies

$$\bar{\delta} \tilde{T}_i^a = \bar{\delta} t T_i^a + t \bar{\delta} T_i^a = \tilde{T}_{,\mu}^a \epsilon^\mu + N^b \epsilon_{,b}^0 \tilde{T}_i^a + \epsilon_{,b}^b \tilde{T}_i^a - N^a \tilde{T}_i^c \epsilon_{,c}^0 - \epsilon_{,c}^a \tilde{T}_i^c. \tag{41}$$

Finally, we also find that

$$
\begin{aligned}
\underset{\sim}{\bar{\delta} N} &= -\underset{\sim}{N} \left( \frac{1}{2} t_a^i \tilde{T}_{i,\mu}^a \epsilon^\mu + \epsilon_{,a}^a + \epsilon_{,a}^0 N^a \right) \\
&\quad + \underset{\sim}{N} \epsilon_{,0}^0 - \underset{\sim}{N} N^a \epsilon_{,a}^0 + \left( \frac{1}{2} t_a^i \tilde{T}_{i,0}^a \underset{\sim}{N} + \underset{\sim}{N}_{,0} \right) \epsilon^0 + \left( \frac{1}{2} t_a^i \tilde{T}_{i,b}^a \underset{\sim}{N} + \underset{\sim}{N}_{,b} \right) \epsilon^b.
\end{aligned} \tag{42}
$$

As noted originally in [16] with regard to Hilbert action, the variations of the lapse and shift are not projectable under the Legendre transformation to phase space due to the dependence on their time derivatives, and the unique means of eliminating these terms in spacetime diffeomorphisms is to require a metric dependence which we rewrite in the form $\tilde{n}^\mu \underset{\sim}{\zeta}^0$, where

$$\tilde{n}^\mu := t n^\mu = \left( \underset{\sim}{N} \right)^{-1} \left( \delta_0^\mu - \delta_a^\mu N^a \right). \tag{43}$$

The general infinitesimal spacetime coordinate variation is therefore

$$\epsilon^\mu = t n^\mu \underset{\sim}{\zeta}^0 + \delta_a^\mu \zeta^a. \tag{44}$$

It should be noted here that this requirement results in a loss of the original spacetime diffeomorphism Lie algebra. The most striking change is a forced dependence on the underlying spatial metric, leading to what has become known as the Bergmann Komar group. A detailed history of this development can be found in [6,19].

Taking this required metric dependence into account, the resulting variations are

$$
\begin{aligned}
\bar{\delta} N &= \underset{\sim}{\dot{\zeta}}^0 - N^a \underset{\sim}{\zeta}_{,a}^0 + \zeta^a N_{,a} = \left( t \underset{\sim}{\zeta}^0 \right)_{,0} - N^a \left( t \underset{\sim}{\zeta}^0 \right)_{,a} + \zeta^a N_{,a} \\
&= t_{,0} \underset{\sim}{\zeta}^0 + t \underset{\sim}{\zeta}_{,0}^0 - N^a t_{,a} \underset{\sim}{\zeta}^0 - N^a t \underset{\sim}{\zeta}_{,a}^0 + \zeta^a N_{,a};
\end{aligned} \tag{45}
$$

therefore,

$$
\begin{aligned}
\underset{\sim}{\bar{\delta} N} &= \bar{\delta} t^{-1} N + t^{-1} \bar{\delta} N \\
&= -t^{-2} \bar{\delta} t N + t^{-1} \left( t_{,0} \underset{\sim}{\zeta}^0 + t \underset{\sim}{\zeta}_{,0}^0 - N^a t_{,a} \underset{\sim}{\zeta}^0 - N^a t \underset{\sim}{\zeta}_{,a}^0 + \zeta^a N_{,a} \right) \\
&= -t^{-2} \bar{\delta} t N + t^{-1} \left( t t_{a,0}^i T_i^a \underset{\sim}{\zeta}^0 + t \underset{\sim}{\zeta}_{,0}^0 - N^a t t_{b,a}^i T_i^b \underset{\sim}{\zeta}^0 - N^a t \underset{\sim}{\zeta}_{,a}^0 + \zeta^a N_{,a} \right).
\end{aligned} \tag{46}
$$

To continue, we need

$$
\begin{aligned}
\bar{\delta} t_a^i &= t_{a,\mu}^i \epsilon^\mu + t_b^i N^b \epsilon_{,a}^0 + t_b^i \epsilon_{,a}^b \\
&= N^{-1} t_{a,0}^i \xi^0 - N^{-1} t_{a,b}^i N^b \xi^0 + t_{a,b}^i \xi^b + t_b^i N^b \left( N^{-1} \xi^0 \right)_{,a} + t_b^i \left( -N^{-1} N^b \xi^0 + \xi^b \right)_{,a} \\
&= N^{-1} t_{a,0}^i \xi^0 - N^{-1} t_{a,b}^i N^b \xi^0 + t_{a,b}^i \xi^b + t_b^i \left( -N^{-1} N_{,a}^b \xi^0 + \xi_{,a}^b \right).
\end{aligned}
\tag{47}
$$

We use this to calculate

$$
-t^{-2} N \bar{\delta} t = -t^{-1} N \bar{\delta} t_a^i T_i^a = -t^{-1} T_i^a \left( t_{a,0}^i \xi^0 - t_{a,b}^i N^b \xi^0 + N t_{a,b}^i \xi^b + t_b^i \left( -N_{,a}^b \xi^0 + N \xi_{,a}^b \right) \right).
\tag{48}
$$

Combining terms, we obtain

$$
\begin{aligned}
\underset{\sim}{\bar{\delta} N} &= -t^{-1} T_i^a \left( t_{a,0}^i \xi^0 - t_{a,b}^i N^b \xi^0 + N t_{a,b}^i \xi^b + t_b^i \left( -N_{,a}^b \xi^0 + N \xi_{,a}^b \right) \right) \\
&\quad + t^{-1} \left( t t_{a,0}^i T_i^a \underset{\sim}{\xi^0} + t \underset{\sim}{\xi_{,0}^0} - N^a t t_{b,a}^i T_i^b \underset{\sim}{\xi^0} - N^a t \underset{\sim}{\xi_{,a}^0} + \xi^a N_{,a} \right) \\
&= -t^{-1} T_i^a \left( N t_{a,b}^i \xi^b + t_b^i \left( -N_{,a}^b \xi^0 + N \xi_{,a}^b \right) \right) \\
&\quad + t^{-1} \left( t \underset{\sim}{\xi_{,0}^0} - N^a t \underset{\sim}{\xi_{,a}^0} + \xi^a N_{,a} \right) \\
&= -\underset{\sim}{N} T_i^a t_{a,b}^i \xi^b + N_{,a}^a \underset{\sim}{\xi^0} - \underset{\sim}{N} \xi_{,a}^a + \underset{\sim}{\xi_{,0}^0} - N^a \underset{\sim}{\xi_{,a}^0} + t^{-1} N_{,a} \xi^a \\
&= N_{,a}^a \underset{\sim}{\xi^0} - \underset{\sim}{N} \xi_{,a}^a + \underset{\sim}{\xi_{,0}^0} - N^a \underset{\sim}{\xi_{,a}^0} + \underset{\sim}{N}_{,a} \xi^a.
\end{aligned}
\tag{49}
$$

Next, we need

$$
\begin{aligned}
\bar{\delta} N^a &= \xi_{,0}^a - N e^{ab} \xi_{,b}^0 + N_{,b} e^{ab} \xi^0 + N_{,b}^a \xi^b - N^b \xi_{,b}^a \\
&= \xi_{,0}^a - N e^{ab} \left( t \underset{\sim}{\xi^0} \right)_{,b} + \left( t \underset{\sim}{N} \right)_{,b} e^{ab} t \underset{\sim}{\xi^0} + N_{,b}^a \xi^b - N^b \xi_{,b}^a \\
&= \xi_{,0}^a - t^2 \underset{\sim}{N} e^{ab} \underset{\sim}{\xi_{,b}^0} + t^2 \underset{\sim}{N}_{,b} e^{ab} \underset{\sim}{\xi^0} + N_{,b}^a \xi^b - N^b \xi_{,b}^a.
\end{aligned}
\tag{50}
$$

As a final step, we need to consider the variations under $\epsilon^\mu = \delta_a^\mu \xi^a$. These contribute the additional terms to the Noether density

$$
p_a^i \tilde{T}_{i,b}^a \xi^b + p_a^i \left( \xi_{,b}^b \tilde{T}_i^a - \epsilon_{,c}^a \tilde{T}_i^c \right).
\tag{51}
$$

After performing an integration by parts, letting $\xi^a \to 0$ as $x^a \to \infty$, we obtain contribution

$$
\begin{aligned}
& -p_a^i \tilde{T}_{i,b}^a \xi^b + \left( p_a^i \tilde{T}_i^a \right)_{,b} \xi^b - \left( p_b^i \tilde{T}_i^a \right)_{,a} \xi^b = \left( p_{a,b}^i \tilde{T}_i^a - p_{b,a}^i \tilde{T}_i^a - p_b^i \tilde{T}_{i,a}^a \right) \xi^b \\
&= -2 D_{[a} p_{b]}^i \tilde{T}_i^a \xi^b =: \tilde{\mathcal{H}}_b \xi^b.
\end{aligned}
\tag{52}
$$

Indeed, since $\xi^a$ is an arbitrary spacetime function, this delivers an additional vanishing Noether generator of spatial diffeomorphisms with constraint

$$
0 = \tilde{\mathcal{H}}_a = -2 \tilde{T}_i^b D_{[b} p_{a]}^i.
\tag{53}
$$

Substituting the original variations into the Noether charge, we obtain

$$
\begin{aligned}
\mathfrak{C}_\epsilon &= p_a^i \tilde{T}_{i,0}^a \epsilon^0 - \mathcal{L}\epsilon^0 \\
&+ p_a^i \tilde{T}_{i,b}^a \epsilon^b + p_a^i\left(N^b \epsilon_{,b}^0 \tilde{T}_i^a + \epsilon_{,b}^b \tilde{T}_i^a - N^a \tilde{T}_i^c \epsilon_{,c}^0 - \epsilon_{,c}^a \tilde{T}_i^c\right) + \tilde{\tilde{P}}\delta\underset{\sim}{N} + \tilde{P}_a \bar{\delta} N^a \\
&= \tilde{\tilde{\mathcal{H}}}_0 \epsilon^0 + \left(p_a^i \tilde{T}_i^a N_{|c}^c - \frac{1}{2}p_{a,b}^i t_b^i \tilde{T}_j^c \tilde{T}_j^a N_{|c}^b - \frac{1}{2}p_b^i \tilde{T}_i^c N_{|c}^b\right)\epsilon^0 \\
&+ p_a^i \tilde{T}_{i,b}^a \epsilon^b + p_a^i\left(N^b \epsilon_{,b}^0 \tilde{T}_i^a + \epsilon_{,b}^b \tilde{T}_i^a - N^a \tilde{T}_i^c \epsilon_{,c}^0 - \epsilon_{,c}^a \tilde{T}_i^c\right) + \tilde{\tilde{P}}\bar{\delta}\underset{\sim}{N} + \tilde{P}_a \bar{\delta} N^a. \quad (54)
\end{aligned}
$$

Next, we collect terms (54) involving $\epsilon^0$ and not $\tilde{\tilde{\mathcal{H}}}_0$. We have

$$
\begin{aligned}
&\frac{1}{2}\left(p_a^i T_i^a e^{cd} - p_a^i T_i^d e^{ac}\right) t g_{e(c} N_{|d)}^e \epsilon^0 + p_a^i\left(N^b \epsilon_{,b}^0 \tilde{T}_i^a - N^a \tilde{T}_i^c \epsilon_{,c}^0\right) \\
&= \frac{1}{2\underset{\sim}{N}}\left(p_a^i \tilde{T}_i^a e^{cd} - p_a^i \tilde{T}_i^d e^{ac}\right) N_{(c|d)}\underset{\sim}{\xi}^0 \\
&- \frac{1}{\underset{\sim}{N}}p_a^i\left(-\frac{1}{\underset{\sim}{N}}N_{,b}N^b \underset{\sim}{\xi}_{,b}^0 \tilde{T}_i^a + N^b \underset{\sim}{\xi}_{,b}^0 \tilde{T}_i^a + \frac{1}{\underset{\sim}{N}}N_{,c}N^a \tilde{T}_i^c \underset{\sim}{\xi}^0 - N^a \tilde{T}_i^c \underset{\sim}{\xi}_{,c}^0\right). \quad (55)
\end{aligned}
$$

We perform an integration by parts in the first line to obtain

$$
\begin{aligned}
&-\frac{1}{2}\left[\underset{\sim}{N}\underset{\sim}{\xi}^0\left(p_a^i \tilde{T}_i^a e^{cd} - p_a^i \tilde{T}_i^{(d}e^{c)a)}\right)\right]_{|d} N_c \\
&= -\frac{1}{2}\left(\underset{\sim}{N}\underset{\sim}{\xi}^0\right)_{|d}\left(p_a^i \tilde{T}_i^a N^d - p_a^i \tilde{T}_i^{(d}N^{a)}\right) \\
&- \frac{1}{2}\underset{\sim}{N}\underset{\sim}{\xi}^0\left(p_{a|d}^i \tilde{T}_i^a N^d - p_{a|d}^i \tilde{T}_i^{(d}N^{a)}\right). \quad (56)
\end{aligned}
$$

In addition, we have

$$
\begin{aligned}
&p_a^i \tilde{T}_{i,b}^a \epsilon^b + p_a^i\left(\epsilon_{,b}^b \tilde{T}_i^a - \epsilon_{,c}^a \tilde{T}_i^c\right) \\
&= p_a^i \tilde{T}_{i,b}^a \underset{\sim}{N}^{-1}N^b \underset{\sim}{\xi}^0 - p_a^i \tilde{T}_i^a\left(-\underset{\sim}{N}^{-2}N_{,b}N^b \underset{\sim}{\xi}^0 + \underset{\sim}{N}^{-1}N_{,b}^b \underset{\sim}{\xi}^0 + \underset{\sim}{N}^{-1}N^b \underset{\sim}{\xi}_{,b}^0\right) \\
&+ p_a^i \tilde{T}_i^b\left(-\underset{\sim}{N}^{-2}N_{,b}N^a \underset{\sim}{\xi}^0 + \underset{\sim}{N}^{-1}N_{,b}^a \underset{\sim}{\xi}^0 + \underset{\sim}{N}^{-1}N^a \underset{\sim}{\xi}_{,b}^0\right). \quad (57)
\end{aligned}
$$

Then, it turns out that some amazing cancelations occur, and the resulting Noether charge is

$$
\begin{aligned}
C_\xi &= \int d^3x \Bigg[\tilde{\tilde{\mathcal{H}}}_0 \underset{\sim}{\xi}^0 + \tilde{\mathcal{H}}_a \xi^a \\
&+ \tilde{\tilde{P}}\left(N_{,a}^a \underset{\sim}{\xi}^0 - \underset{\sim}{N}\underset{\sim}{\xi}_{,a}^a + \underset{\sim}{\xi}_{,0}^0 - N^a \underset{\sim}{\xi}_{,a}^0 + N_{,a}\xi^a\right) \\
&+ \tilde{P}_a\left(\xi_{,0}^a - t^2 \underset{\sim}{N}e^{ab}\underset{\sim}{\xi}_{,b}^0 + t^2 N_{,b}e^{ab}\underset{\sim}{\xi}^0 + N_{,b}^a \xi^b - N^b \xi_{,b}^a\right)\Bigg], \quad (58)
\end{aligned}
$$

where we have the additional vanishing constraint due to the arbitrariness in function $\underset{\sim}{\xi}^0$,

$$
\tilde{\tilde{\mathcal{H}}}_0 = -\frac{1}{2}\tilde{T}_i^a \tilde{T}_j^b\left({}^3R_{ab}^{ij} + p_a^i p_b^i - p_b^i p_a^j\right) = 0. \quad (59)
$$

Similarly, since $\xi^a$ can vary arbitrarily in time, we obtain constraint

$$\mathcal{H}_a = 0. \tag{60}$$

These results imply, of course, that $C_{\tilde\xi}$ itself vanishes. [2]

## 4. Spacetime Diffeomorphism-Related Noether Generator

We work out here the requirement to add gauge transformations to the diffeomorphisms in order to attain projectability under the Legendre transformation from configuration–velocity space to phase space. This challenge arises due to the absence of anti-symmetrized linear combinations of triad time derivatives in the ADM Lagrangian. This is a combination that appears in the Ricci rotation coefficient (see [20]).

$$\Omega_0^{ij} = -\tilde{T}_{,0}^{a[i}\,\underset{\sim}{t}_a^{j]} - N_{,b}^a t_a^{[i}T^{j]b} + N^c t_c^k T^{a[i}T^{j]b}t_{a,b}^k + N^c t_{c,b}^{[i}T^{j]b}. \tag{61}$$

We undertake the variation of the covector component $\Omega_0^{ij}$ under the infinitesimal diffeomorphism with descriptor $\epsilon^\mu = n^\mu \xi^0 + \delta_a^\mu \xi^a$,

$$\bar\delta\Omega_0^{ij} = \Omega_\mu^{ij}\epsilon_{,0}^\mu + \delta\Omega_0^{ij}. \tag{62}$$

We do not need $\delta\Omega_0^{ij}$ since it is projectible. Thus, we have

$$\bar\delta\Omega_0^{ij} = \Omega_0^{ij}\left(N^{-1}\xi^0\right)_{,0} + \Omega_a^{ij}\left(-N^{-1}N^a\xi^0 + \xi^a\right)_{,0} + \dots. \tag{63}$$

We discover that the unprojectable time derivatives of the lapse and shift appear in this variation. But the good news is that these inadmissible variations can be eliminated by adding gauge rotations with

$$\eta^k = -\epsilon^{kij}\Omega_\mu^{ij}n^\mu\xi^0, \tag{64}$$

with generator

$$-\int d^3x \,\epsilon^{kij}\Omega_\mu^{ij}n^\mu\xi^0 p_k = -\int d^3x \,\epsilon^{kij}\Omega_\mu^{ij}n^\mu\xi^0\epsilon^{kmn}p_a^m\tilde{T}_a^a$$
$$= \int d^3x \,\Omega_\mu^{k[i}\,\underset{\sim}{t}_a^{j]}n^\mu\tilde{T}_k^a\underset{\sim}{\xi}^0. \tag{65}$$

The additional Ricci rotation coefficient is (from [20]) the three-dimensional coefficient $\Omega_a^{ij} = \omega_a^{ij}$.

Adding this expression to the first line in (58), we define the vanishing generator density

$$\mathcal{H}_0 := \left(-{}^3R + \frac{1}{4}p_a^i p_b^i e^{ab} - \frac{1}{4}p_a^i T_i^a p_b^j T_j^b + \Omega_\mu^{k[i}\,\underset{\sim}{t}_a^{j]}n^\mu\tilde{T}_k^a\right) = 0. \tag{66}$$

Thus, we finally have the full diffeomorphism-related vanishing Noether generator, derived directly from the vanishing Noether charge,

$$\begin{aligned}
C_{\xi\eta} \;=\; & \int d^3x \Bigg[ \mathcal{H}_0\underset{\sim}{\xi}^0 + \mathcal{H}_a\xi^a + \eta^k\mathcal{H}_k \\
& + \; \tilde{\tilde{P}}\left( N_{,a}^a\underset{\sim}{\xi}^0 - \underset{\sim}{N}\xi_{,a}^a + \underset{\sim}{\xi}_{,0}^0 - N^a\underset{\sim}{\xi}_{,a}^0 + \underset{\sim}{N}_{,a}\xi^a \right) \\
& + \; \tilde{P}_a\left( \xi_{,0}^a - t^2 N e^{ab}\underset{\sim}{\xi}_{,b}^0 + t^2\underset{\sim}{N}_{,b}e^{ab}\underset{\sim}{\xi}^0 + N_{,b}^a\xi^b - N^b\xi_{,b}^a \right) \Bigg]. 
\end{aligned} \tag{67}$$

## 5. Variations Produced by the Generators and the Generator Algebra

We first confirm here that the Noether generators that were obtained do indeed generate the correct variations of configuration variables. (One could indeed invoke a procedure invented by Rosenfeld in 1930 to show that momentum variables also undergo the correct variations.) We begin with the variations generated by $R(\xi) := \int d^3x \mathcal{H}_k \xi^k$. We have

$$\delta_\xi \widetilde{T}_i^a = \left\{ \widetilde{T}_i^a, R(\xi) \right\} = -\epsilon^{ijk} \xi^j \widetilde{T}_k^a. \tag{68}$$

Next, we consider variations generated by $V_{\vec{\eta}} := \int d^3x \mathcal{H}_a \eta^a$, finding that

$$\delta_{\vec{\eta}} \widetilde{T}_i^a = \left\{ \widetilde{T}_i^a, V(\vec{\eta}) \right\} = -\widetilde{T}_i^b \eta^a_{,b} + \widetilde{T}_{i,b}^b \eta^a - \eta^b \omega_b^{ji} \widetilde{T}_j^a. \tag{69}$$

Here, it is useful to define $\omega_b^m := \frac{1}{2} \epsilon^{mjk} \omega_b^{jk}$, from which it follows that $\omega_b^{ji} = \epsilon^{jik} \omega_b^k$. Thus, we can interpret the last term in (69) as arising from a triad gauge rotation with the gauge descriptor derived from the spatial diffeomorphism descriptor, i.e., the full variation becomes such that the first is a spatial diffeomorphism variation and the second is a gauge rotation,

$$\delta_{\vec{\eta}} \widetilde{T}_i^a = -\widetilde{T}_i^b \eta^a_{,b} + \widetilde{T}_{i,b}^b \eta^a - \epsilon^{ijk} \eta^b \omega_b^j \widetilde{T}_k^a. \tag{70}$$

The final variation to consider is that generated by $S[\underset{\sim}{\xi}] := \int d^3x \mathcal{H}_0 \underset{\sim}{\xi}^0$. We have

$$\delta_{\underset{\sim}{\xi}} \widetilde{T}_i^a = \left\{ \widetilde{T}_i^a, S[\underset{\sim}{\xi}] \right\} = \left\{ \widetilde{T}_i^a, \left\{ \widetilde{T}_i^a, S[\underset{\sim}{\xi}^0] \right\} \right\} = \frac{1}{2} T_j^b \left( p_b^i T_j^a - p_b^j T_i^a \right) \underset{\sim}{\xi}^0. \tag{71}$$

The vanishing constraint generators that we obtained here are precisely those obtained previously in [20]. Indeed, this algebra played a central role in the derivation of the complete generator of spacetime diffeomorphism related transformations that we derived here in an alternative manner in applying Noether's second theorem. The generators obey the following closed Poisson bracket algebra:

$$\{ R[\xi], R[\eta] \} = -R[[\xi, \eta]], \tag{72}$$

where $[\xi, \eta]^i := \epsilon^{ijk} \xi^j \eta^k$,

$$\left\{ V[\vec{\xi}], R[\eta] \right\} = \left\{ S[\underset{\sim}{\xi}], R[\eta] \right\} = 0, \tag{73}$$

$$\left\{ V[\vec{\xi}], V[\vec{\eta}] \right\} = V[[\vec{\xi}, \vec{\eta}]] - R[^3R_{ab} \xi^a \eta^b], \tag{74}$$

using the Lie bracket

$$[\vec{\xi}, \vec{\eta}]^a := \xi^b \eta^a_{,b} - \eta^b \xi^a_{,b}, \tag{75}$$

$$\left\{ S[\underset{\sim}{\xi}], V[\vec{\eta}] \right\} = -S[\mathcal{L}_{\vec{\eta}} \underset{\sim}{\xi}^0] - R[\eta^a \delta_{PD}[t \underset{\sim}{\xi}^0] + \delta_R[\Omega_\mu n^\mu t \underset{\sim}{\xi}^0] \omega_a], \tag{76}$$

where

$$(\delta_{PD}[t \underset{\sim}{\xi}^0] + \delta_R[\Omega_\mu n^\mu t \underset{\sim}{\xi}^0]) \omega_a^i = (p_a^j \widetilde{T}^{bk} D_b(\underset{\sim}{\xi}^0) + 2\widetilde{T}_j^b \underset{\sim}{\xi}^0 D_{[a} p_{b]}^k) \epsilon^{ijk}, \tag{77}$$

and finally,

$$\left\{ S[\underset{\sim}{\xi}], S[\underset{\sim}{\eta}] \right\} = V[\vec{\zeta}] - R[[D_a \underset{\sim}{\xi}^0 \widetilde{T}^a, D_b \underset{\sim}{\eta}^0 \widetilde{T}^b]], \tag{78}$$

where

$$\zeta^a = \left( \underset{\sim}{\xi}^0 \partial_b \underset{\sim}{\eta} - \underset{\sim}{\eta} \partial_b \underset{\sim}{\xi} \right) \widetilde{\widetilde{e}}^{ab}. \tag{79}$$

## 6. The Canonical Hamiltonian

It must be stressed that the above diffeomorphism generator differs in an essential manner from the conventional temporal evolution generator. Note that the canonical Hamiltonian derived in (23), after performing an integration by parts (assuming $N^a \to 0$ as $\vec{x} \to \infty$), takes the form

$$\mathcal{H}_c = \tilde{\tilde{\mathcal{H}}}_0 + N^a \tilde{\mathcal{H}}_a, \tag{80}$$

and, as noted in (25), the Poisson bracket $\left\{ \tilde{T}_i^a, \int d^3x \mathcal{H}_c \right\}$ yields an equation of motion that does not take into account the spacetime coordinate freedom available in the function $\Omega^k$ that multiplies the triad gauge constraint in the complete Hamiltonian

$$H = \int d^3x \left( \underset{\sim}{N} \tilde{\tilde{\mathcal{H}}}_0 + N^a \tilde{\mathcal{H}}_a + \Omega^k \tilde{\mathcal{H}}_k \right). \tag{81}$$

This evolves initial phase space data in time via the Poisson bracket. The generator $C_{\xi\eta}$, on the other hand, acts on the entire solutions generated by $H$ and transforms them to new physically equivalent solutions that are related through the action of active spacetime diffeomorphisms.

## 7. Extension to the Barbero–Immirzi–Holst Model

The Holst addition to the Lagrangian is

$$\mathcal{L}_H = \frac{1}{4\gamma} N t E_I^\mu E_J^\nu {}^4 R_{\mu\nu}^{IJ}. \tag{82}$$

It is introduced with what has become known as the Barbero–Immirzi parameter $\gamma$. The curvature is expressed in terms of the Ricci rotation coefficients,

$$ {}^4 R_{\mu\nu}^{IJ} = \partial_\mu \Omega_\nu^{IJ} - \partial_\nu \Omega_\mu^{IJ} + \Omega_\mu^{IM} \Omega_{\nu M}{}^J - \Omega_\nu^{IM} \Omega_{\mu M}{}^J. \tag{83}$$

It is of course well known that this Lagrangian vanishes when, as we assume, the torsion vanishes. The outcome for our specific use is that the new canonical momentum $p_a^{\gamma i}$ is obtained through a canonical transformation of $p_a^i$, i.e.,

$$p_a^{\gamma i} = p_a^i + \frac{1}{2} \gamma^{-1} \epsilon^{ijk} \omega_a^{jk}. \tag{84}$$

It follows that we need only make this substitution for $p_a^i$ in our Noether generator (67) to obtain the spacetime diffeomorphism-related symmetry generator in the Barbero–Immirzi–Holst model.

## 8. Evolving Constants of Motion

We briefly overview here the manner in which the vanishing diffeomorphism-related generator may be employed to implement the use of intrinsic coordinates, evoking the general method presented in [21]. There, we proposed the use of intrinsic coordinates which must be spacetime scalar phase space functions. We represent them here as $X^\mu \left( \tilde{T}_i^a, p_j^b \right)$ [3]. With their aid, we can establish gauge conditions which we represent as $\chi^{(1)\mu} = x^\mu - X^\mu = 0$. Recognizing that these must be preserved under time evolution, we obtain a second set of gauge conditions

$$0 = \frac{d}{dt} \chi^\mu = \delta_0^\mu - N^\rho \{ X^\mu, \mathcal{H}_\rho \} = \delta_0^\mu - \mathcal{A}_\rho^\mu N^\rho =: \chi^{(2)\mu}, \tag{85}$$

where

$$\mathcal{A}_\rho^\mu := \{ X^\mu, \mathcal{H}_\rho \}. \tag{86}$$

In [21], we extended a procedure that was invented in [22] so as to include the lapse and shift as phase space variables. The basic idea is to take linear combinations of the eight first-class constraints which we represent here by $\zeta_{(j)\nu} = \left(\mathcal{H}_\mu, \tilde{\tilde{P}}, \tilde{P}_a\right)$, employing the inverse of $\mathcal{A}^\mu_\rho$. Representing the new set of the original first-class constraints by $\bar{\zeta}_{(j),\mu}$, we are able to arrange that they satisfy the Poisson brackets with the gauge conditions satisfying

$$\left\{\chi^{(i)\mu}, \bar{\zeta}_{j,\nu}\right\} = -\delta^i_j \delta^\mu_\nu. \tag{87}$$

Consequently, we can solve for the gauge functions $\bar{\xi}^\mu$ which transform arbitrary solutions of the field equations to those that satisfy the gauge conditions. Of course, in doing so, in this case, we make use of Generator (67) with the new linear combinations of constraints $\zeta_{(j)\nu}$. Thus, for any phase space function $\Phi$, including the lapse and shift, we can construct the corresponding spacetime invariant $\mathcal{I}_\Phi$ through the action of the generator $C_{\bar{\xi}}$, i.e.,

$$\mathcal{I}_\Phi = exp\left(\left\{-, C_{\bar{\xi}}\right\}\right)\Phi. \tag{88}$$

The validity of this expansion has been demonstrated, for example, in [1,21], for several previous models. It will be straightforward to do so for the classical Barbero–Immirzi–Holst theory. A cosmological perturbative approach employing these expansions would be of particular interest.

## 9. Conclusions

We presented here a new direct method for obtaining the generator of spacetime diffeomorphism-related phase space transformations through appealing directly to Noether's second theorem. The question that must now be addressed is how one can take these classical symmetries into account in an eventual quantum theory of gravity. Much effort has of course long been devoted to addressing this issue. Pullin and his collaborators have certainly made significant progress in addressing the associated problem of time [23]. Rovelli has long advocated a closely related approach in which a subset of fields serve as clocks. In this regard, we are choosing Weyl scalars expressed in terms of phase space variables as both temporal and spatial intrinsic coordinates [24]. This is accomplished in a manner as advocated in [1,21].

**Funding:** This research received no external funding.

**Data Availability Statement:** Data are contained within the article.

**Conflicts of Interest:** The author declares no conflict of interest.

## Notes

[1]    A major advantage in employing the ADM Lagrangian is that it does vary as a Lagrangian density, assuming only that variations at spatial infinity vanish. See [17], p. 119 and [18].

[2]    It is likely a surprise to most readers that this procedure for determining what are now known as secondary constraints, following the so-called Bergmann–Dirac procedure, was initiated by Léon Rosenfeld in 1930. We believe it would be more accurate to refer to the Rosenfeld–Bergmann–Dirac method. The relation between Bergmann, Rosenfeld and Dirac is analyzed in detail in [6].

[3]    The analogues have long been represented by several authors as $T^\mu$ and they have been denoted as "clock" variables. See, for example, [8]. We recommend referring to $T^0$ as a clock variable and the $T^a$ rod variables.

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
