# Peer review of "Diffeomorphism Covariance of the Canonical Barbero–Immirzi–Holst Triad Theory"

_universe, doi:10.3390/universe9110458_

Round 1

Reviewer 1 Report

Comments and Suggestions for Authors

Noether’s second theorem is used to obtain the spacetime diffeomorphism-related symmetry generator in the Barbero-Immirzi-Holst model. This is an interesting and new approach to the problem.

Author Response

Thank you so much for your positive review. I have submitted a revised manuscript containing additional material suggested by reviewer 3.

Reviewer 2 Report

Comments and Suggestions for Authors

The paper concerns the analysis of four-dimensional spacetime diffeomorphism symmetry viewed as a canonical symmetry in Hamiltonian formalism. The construction of relevant generators is related to the second Noether theorem. The author deals with the ADM Lagrangian and its Barbero-Immirzi-Holst extension. He provides the very detailed calculations which might be useful for the people involved in the field. Even though the problem of description of canonical, in particular symmetry, transformations in constrained Hamiltonian systems and its relation to the second Noether theorem is rather well understood, such explicit calculations concerning quite complicated system are instructive. Therefore, I recommend the publication.

Author Response

(The authors gave the same response as above.)

Reviewer 3 Report

Comments and Suggestions for Authors

The article deals with the Hamiltonian formalism of the Barbero-Immirz-Holst theory. The Lagrangian is constructed and from there the symmetry group generators are derived using Noether's theorem. The article is generally well written, despite its introduction as a summary. It would be helpful to contextualize the different approaches to Hamiltonian formalism in gravitation. 

Some points should be further clarified:

1. What is the algebra of constraints in (5.1)? Does such an algebra hold in all space or only in an asymptotic limit?

2. What is the class of the constraints?

3. Normally the temporal evolution of a certain quantity is given by the commutator between that quantity and the Hamiltonian, which does not seem to be the case in equation (7.1). A clarification is necessary.

Therefore, I cannot recommend publishing this article in its present form.

Author Response

I have responded to each of the following reviewer comments

  1. What is the algebra of constraints in (5.1)? Does such an algebra hold in all space or only in an asymptotic limit?

The full first class algebra is given in the new section 5. It holds throughout spacetime - with the sole condition, as pointed out - that the shift functions N^a go to zero at spatial infinity.

2. What is the class of the constraints?

The constraints are all first class. The full Poisson bracket algebra is given in section 5.

3. Normally the temporal evolution of a certain quantity is given by the commutator between that quantity and the Hamiltonian, which does not seem to be the case in equation (7.1). A clarification is necessary.

I have clarified this in section 6

Let me also add that I have included some more historical background in the introductory section, with additional references.

Round 2

Reviewer 3 Report

Comments and Suggestions for Authors

The author have changed the manuscript according to my previous report. The manuscript is a better shape now, thus I recommend it to publication as it is.